# Association of Alternative Dietary Patterns with Osteoporosis and Fracture Risk in Older People: A Scoping Review

**DOI:** 10.3390/nu15194255

**Published:** 2023-10-03

**Authors:** Huiyu Chen, Christina Avgerinou

**Affiliations:** 1Division of Medicine, University College London, London WC1E 6BT, UK; huiyu.chen.21@alumni.ucl.ac.uk; 2Department of Primary Care and Population Health, University College London, London NW3 2PF, UK

**Keywords:** ageing, dietary patterns, diet scores, osteoporosis, osteoporotic fractures

## Abstract

Purpose: Although the Mediterranean diet has been associated with a lower risk of hip fracture, the effect of other dietary patterns on bone density and risk of fracture is unknown. This scoping review aims to investigate the association between adherence to alternative dietary patterns (other than the traditional Mediterranean diet) and osteoporosis or osteoporotic fracture risk in older people. Methods: A systematic search was carried out on three electronic databases (Medline, EMBASE, and Scopus) to identify original papers studying the association between alternative dietary patterns (e.g., Baltic Sea Diet (BSD), modified/alternative Mediterranean diet in non-Mediterranean populations, Dietary Approaches to Stop Hypertension (DASH)) assessed using ‘prior’ methods (validated scores) and the risk of osteoporotic fracture or Bone Mineral Density (BMD) in people aged ≥50 (or reported average age of participants ≥ 60). Results from the included studies were presented in a narrative way. Results: Six observational (four prospective cohort and two cross-sectional) studies were included. There was no significant association between BMD and BSD or DASH scores. Higher adherence to DASH was associated with a lower risk of lumbar spine osteoporosis in women in one study, although it was not associated with the risk of hip fracture in another study with men and women. Higher adherence to aMED (alternative Mediterranean diet) was associated with a lower risk of hip fracture in one study, whereas higher adherence to mMED (modified Mediterranean diet) was associated with a lower risk of hip fracture in one study and had no significant result in another study. However, diet scores were heterogeneous across cohort studies. Conclusions: There is some evidence that a modified and alternative Mediterranean diet may reduce the risk of hip fracture, and DASH may improve lumbar spine BMD. Larger cohort studies are needed to validate these findings.

## 1. Introduction

With the ageing trend dramatically increasing, a global focus has been on healthy longevity in older people. Ageing tends to affect the nutritional status of older people. As people age, their energy requirements and food intake will dramatically reduce, which can lead to malnutrition [1]. Combined with the impact of ageing on hormone levels and bone health, these adverse factors can lead to low Bone Mineral Density (BMD) and increased incidence of osteoporotic fractures in older people [2,3]. Hence, osteoporosis is one of the leading causes of morbidity and mortality in older people, and its prevalence ranks the highest among non-communicable diseases worldwide, at about 0.83% [4]. It is estimated that more than 200 million people have osteoporosis, and 137 million women and 21 million men over the age of 50 are at or near the fracture threshold globally, with an average of 8.9 million fractures occurring annually, and this figure is expected to double by 2040 [5,6].

Age, gender, region, and race are main risk factors for osteoporosis. Specifically, a person’s bone mass declines in adulthood (25–30 years) at a rate of 0.5% per year, while bone loss occurs earlier in women than in men, with a rapid decline in bone mass occurring at ages 65 to 69 for women and 74 to 79 for men [7,8]. Notably, the mortality-adjusted lifetime fracture risk for 60-year-old women is about 44%, nearly double that of men of the same age (25%) [9]. Additionally, the incidence of osteoporosis (without gender difference) increases progressively from tropical countries (<150/100,000 cases per year) to northern temperate countries (>250/100,000 cases per year) [10]. Some studies show that African American women’s BMD is consistently higher than that of white women and that African Americans and Asians have shorter hip axis lengths than other ethnic groups, which may be another explanation for their reduced fracture risk [11,12]. 

Although osteoporosis is a preventable and treatable disease, it has a long incubation period and is often not detected until a fracture occurs. Frailty can affect older people’s physical, psychological, and social functioning as well as their quality of life, and it also increases the risk of falling, making them prone to fractures [13]. Osteoporotic fractures are often followed by surgery and lengthy rehabilitation, which pose a considerable public health burden regarding societal expenditures and health status [14,15]. Shockingly, excluding the indirect costs such as lost productivity and disability, the direct costs associated with treating osteoporotic fractures are estimated to be between $500 billion and $6500 billion annually in Europe, Canada, and the United States alone [16]. Because this figure will continue to rise as the ageing population increases, national healthcare systems should prioritise providing preventive measures for osteoporosis to reduce the number of patients with osteoporotic fractures and reduce costs at the source. Considering the physical characteristics of older people, developing accessible programs that encourage lifestyle modifications, notably balanced and nutritional food intake, is essential to reduce their risk of osteoporosis [17]. 

Programmes for the prevention and treatment of osteoporosis can easily focus on diet as it is a modifiable risk factor for chronic disease. Some small-scale studies found that higher consumption of fish and shellfish has a significant relationship with increased bone mass and reduced osteoporosis risk and higher intake of fruit and vegetables also contributes to lower risk of osteoporosis [18,19]. However, the effect of isolated foods and nutrients is controversial; for example, although we know that sufficient calcium intake is necessary to prevent bone loss, the evidence regarding dairy product consumption and the risk of osteoporotic fracture is conflicting [20,21]. Considering the complex interaction between foods and nutrients and that the impact of diet on people is long-lasting, there has been increasing research interest in adherence to specific dietary patterns and the benefits or risks these might entail with regards to the occurrence of various conditions. 

The biggest challenge in studying the association between dietary patterns and disease risk is to develop a quantitative method to identify dietary patterns unless specific patterns have been already identified [22]. There are two statistical approaches to assess the impact of dietary patterns: data-driven (‘a priori’ methods) and investigator-driven (‘posterior’ methods) [23]. On the one hand, ‘a priori’ methods use a dietary quality scoring system (such as the Baltic Sea Diet (BSD) score, or Alternate Healthy Eating Index 2010 (AHEI) score, etc.) to assess how well the populations adhered to pre-set eating patterns. These methods can describe overall dietary characteristics and are used to predict illness among persons by evaluating the foods or nutrients ingested by individuals and summarising the results to obtain dietary quality scores [23]. On the other hand, ‘posterior’ methods analyse eating patterns derived empirically from observed dietary intake using exploratory factor analysis (EFA), principal component analysis (PCA), or cluster analysis [24]. These methods derive dietary patterns by grouping highly related food groups using data simplification methods and using subjective criteria to determine the amount of retained dietary patterns [25]. It is worth noting that relatively few epidemiological studies have successfully studied the link between dietary patterns and osteoporotic fracture, and ‘posterior’ approaches cannot accurately reflect adherence to a specific diet. 

A popular alternative dietary pattern, the Mediterranean diet (MD), has been recently shown to have a positive impact on musculoskeletal health and reduced incidence of hip fractures [26,27,28]. Higher adherence to the MD has also been associated with an increase in calcium intake in peri- and postmenopausal women [29]. According to a study, olive oil in the MD is rich in olive polyphenols, which are effective in reducing oxidative stress and inflammation, and regular consumption of olive oil can prevent bone loss and improve bone loss markers [30]. Additionally, the MD promotes the intake of n-3 polyunsaturated fatty acids (n-3 PUFA) and vitamin D (abundant in fish), which are positively associated with BMD [18]. Notably, current studies always use traditional MD scores to assess its influence on the risk of osteoporotic fracture in Mediterranean populations. However, the ingredients of the traditional MD (e.g., olive oil) are not always available in other geographical locations [28]. Hence, the modified Mediterranean diet (mMED) score or alternative Mediterranean diet (aMED) score is more accurate in evaluating adherence to the MD in people residing outside the coastal Mediterranean region. 

At the same time, there is also evidence that a pro-inflammatory diet increases the risk of osteoporosis and fracture [31]. To quantify the risk of inflammation, many studies have used the Dietary Inflammatory Index (DII) to provide a total score of the inflammatory potential of nutrients and foods in the dietary pattern, concluding that in order to reduce the incidence of osteoporotic fractures in older adults, the DII in the dietary pattern should be reduced [32,33,34]. 

Notably, there are two previous systematic reviews using an existing ‘a priori’ dietary index, which demonstrated that higher adherence to the traditional MD dietary pattern (based on a traditional MD score) is associated with a reduced risk of hip fracture [27] and that a diet with a high inflammatory potential (manifested by a higher DII score) is associated with increased risk of osteoporosis and fracture [33]. However, there are fewer studies that evaluated the effect of adherence to an alternative dietary pattern such as the Nordic diet, Dietary Approaches to Stop Hypertension (DASH) diet or Chinese diet on bone density and osteoporosis/risk of fracture in older people using other scoring systems (such as the BSD score, AHEI score, or DASH score, etc.). A scoping review of the evidence on this topic can shed light on the role of following a specific diet with regards to bone health. If one or more traditional dietary patterns are found to be protective against osteoporosis, this information could lead to further research to establish these associations and could eventually help inform guidelines for healthy eating to maintain bone density in older adults, especially for non-Mediterranean populations.

This scoping review will seek evidence from observational studies assessing the association between various alternative dietary patterns and BMD or the risk of osteoporotic fractures in people aged over 50 years old by using ‘a priori’ diet scores (excluding traditional MD or DII scores that have been studied previously).

## 2. Methodology

### 2.1. Search Strategy

A comprehensive literature search was conducted for this scoping review, and the whole article was completed in accordance with Preferred Reporting Items for Systematic Reviews and Meta-Analyses for Scoping Reviews (PRISMA-ScR) guidelines [35]. Specifically, references were obtained from Medline (via Ovid), Embase (via Ovid), and Scopus (via UCL library), searching for articles published from 2000 to 22 June 2022 to identify all the original papers that assess the relationship between various alternative dietary patterns/scores and BMD, osteoporosis risk, or osteoporotic fracture risk in older people (see Medline search strategy in Appendix A). 

### 2.2. Screening

The literature screening for this review set the following requirements (protocol available upon request): 

Inclusion Criteria
Study design


Journal articles reporting results of studies conducted on humans published between 2000 and June 2022 in the English language;

Observational studies using a prospective cohort or cross-sectional study design. 

Participants

People aged 50 years and above or, when the age range was not reported, the average age of participants was ≥ 60 years;

The study group was not limited by race or ethnicity.

Exposure

Alternative dietary patterns should be derived by an ‘a priori’ method (using validated dietary indices and scores, such as the Nordic Diet score, alternate Mediterranean diet (aMED) score, modified-Mediterranean Diet Score or other validated tool in non-Mediterranean populations, Healthy Eating Index (HEI), Alternate Healthy Eating Index 2010 (AHEI) or DASH diet score, etc.).

Outcomes

All results to determine the effect of alternative dietary patterns (or adherence to ‘a priori’ diet scores) on the risk of osteoporotic fracture or low Bone Mineral Density.

Exclusion Criteria
Conference abstracts, reviews or editorials;Studies evaluating the effect of interventions, e.g., Randomised Controlled Trials;Studies using ‘posterior’ methods to evaluate the diet pattern;Studies evaluating the effect of the traditional Mediterranean diet in Mediterranean populations (as this has been studied elsewhere). If a study reported results involving a mixture of populations, only those relevant to the non-Mediterranean populations were included;Studies assessing the effect of dietary patterns on osteoporotic fractures by DII (its application has been widely studied).

When a reference was excluded, the following fixed hierarchy was used to identify the exclusion reason to minimise potential conflicts: (1) full text not available, (2) wrong type of publication (e.g., conference abstracts or conference proceeding), (3) wrong study type (e.g., randomized controlled trials or studies testing interventions based on specific nutrients or foods), (4) wrong population (e.g., not fulfilling the age criterion), and (5) wrong investigation methods (e.g., using ‘posterior’ methods). Titles and abstracts were screened by the first author (HC) only, whereas full-text papers were screened by both authors (HC and CA).

### 2.3. Data Extraction

For all full texts cited in the analysis, the first author (HC) extracted the data using a pre-defined data extraction form. The extracted data comprised bibliographic information (first author, country and year of publication), sample size, participant characteristics (gender and age), study design, dietary patterns, diet scores (scoring component, scoring method and criteria), and main findings.

### 2.4. Descriptive Synthesis

The characteristics of research methods (including diet scores), the impact of adherence to different dietary patterns or diet scores on BMD, osteoporosis, and the risk of osteoporotic fracture in various body parts were summarized and compared by both authors (HC and CA). Considering various potential influencing factors that might have an impact on the outcomes, results of a multivariable-adjusted analysis were presented (comparing quantiles of diet scores).

Results were identified as primary outcomes in the paper when: (1) they were the main findings mentioned in the article, or (2) multifactor adjusted results in the BMD/osteoporosis/osteoporotic fracture analysis were included for these outcomes. If neither criterion 1 nor 2 could be applied, all outcomes were taken into consideration as primary outcomes.

## 3. Results

### 3.1. Study Selection

The PRISMA flowchart of the article selection process is presented in Figure 1. The searches retrieved 507 titles and abstracts in total. At the title and abstract screening stage, 180 articles were eliminated because of duplication and 286 due to irrelevancy. Of the remaining forty-one full-text papers that were selected for screening, thirty-five were excluded for various reasons, including four due to publication type, three due to study type, four due to population, and twenty-four due to an unmatched research method. Finally, six articles that met the standards were included in this review.

### 3.2. Characteristics of Included Studies

The general characteristics of the six observational studies assessing the relationship between various alternative dietary patterns/diet scores and BMD, osteoporosis and fractures are shown in Table 1. All of the selected studies were published after 2015, and they were all conducted in non-Mediterranean countries (Europe (50%), America (50%), and Asia (16.7%)). Four were prospective cohort studies, and two were cross-sectional studies. Regarding the study population, female participants were recruited in all six studies, whereas only three studies were conducted in both females and males. The age range of study subjects ranged from 50 to 85 years. The sizes of the study population varied greatly: 50% > 10,000, 16.7% between 1000 and 10,000, and 33.3% < 1000 people.

The assessment of dietary patterns was performed using diet indexes (more information about these is provided in the section below). More specifically, the BSD score [36], DASH diet score [37], HEI-2010 score [38], and AHEI-2010 score [38,39] were used to measure adherence to the BSD, DASH diet and Healthy Eating dietary patterns, respectively. The mMED [40,41] and aMED [38,39] scores were used to measure adherence to the traditional MD for non-Mediterranean populations.

Notably, a study that assessed the relationship between the MEDI-LITE (literature-derived Mediterranean diet) score and bone health was not included in the analysis due to high similarity to the traditional MD score [42]. Furthermore, one of the eligible studies included a pooled analysis from different cohorts, and in this case, we only report findings relevant to the non-Mediterranean populations (Sweden and USA), excluding Greece, where the population has access to the traditional MD [41].

In terms of outcomes, two studies examined BMD [36,37]. One study recorded incident hip fractures and other osteoporotic fractures using data from the Swedish National Patient Register that were linked to the individual via a personal identification number [40]. One study assessed hip fracture using self-reported data from a questionnaire [39]. In another study, all fracture outcomes were self-reported except hip fractures, which were assigned a diagnosis by locally trained staff and centrally confirmed by a second medical record review [38]. Finally, in another study, information on incident hip fractures was collected through local or national patient registers and individual linkage with personal identification numbers in Sweden and using self-reported information from questionnaires or telephone interviews in the USA [41].

**Table 1 nutrients-15-04255-t001:** Summary of main study findings.

First Author, Year	Country (No.)	Gender (F/M) and Population	Mean Age (Age Range) (Years)	Study Design	Dietary Pattern	Diet Score	Outcome	Main Associations Studies	Statistical Measure of Effect
Erkkilä et al., 2017 [36]	Finland	F 554	67.9 (65–71)	Prospective cohort	BSD	BSD score	Bone Mineral Density (BMD)	Association between quartiles of BSD (Q1-Q2-Q3-Q4) score and BMD (Femoral, lumbar, or total body).	Lumbar BMD: *p* = 0.428 (NS) *
									Femoral neck BMD: *p* = 446 (NS) *
									Total body BMD: *p* = 0.294 (NS) *
Shahriarpour et al., 2020 [37]	Iran	F 151	61.2 (50–85)	Cross sectional	DASH diet	DASH score	Bone Mineral Density (BMD)	1. Association between tertiles of DASH score (T1-T2-T3) and BMD (Femoral neck or lumbar spine).	Lumbar spine BMD overall difference across tertiles: *p* = 0.068 ** (NS)
									Lumbar spine BMD; pairwise difference between T3-T1: *p* = 0.594 ** (NS)
									Femoral neck BMD overall difference across tertiles: *p* = 0.323 ** (NS)
									Femoral neck BMD pairwise difference between T3-T1: *p* = 0.921 ** (NS)
									Lumbar spine osteoporosis: OR = 0.28 (95% CI 0.09–0.88) (*p* = 0.029) **
								2. Association between adherence to the DASH dietary pattern in different tertile divisions (T1-T2-T3) and risk of osteoporosis	Femoral neck osteoporosis: OR = 1.21 (95%CI 0.21–7.00) ** (NS)
Byberg et al., 2016 [40]	Sweden	Total 71,306 F (33,403) M (37,903)	60	Cross sectional	Mediterranean-like diet	mMED score	First incident hip fracture (main outcome)	Association between adherence to the mMED score in different tertile divisions (T1-T2-T3) and the incidence of osteoporotic fracture in both genders.	Comparing the highest quintile of adherence to the mMED (6 to 8 points) with the lowest (0 to 2 points):
							First incident fracture of any type and first incident non-hip fracture (secondary outcomes) -		Both genders: hip fracture risk: HR = 0.78 (95%CI 0.69–0.89) *****
Haring et al., 2016 [38]	The United States	F 7916	63.6 (63.6 ± 7.4)	Prospective cohort	No prescribed eating patterns	aMED score, HEI-2010 score, AHEI-2010 score, DASH score	Incident hip and total fractures	Association between adherence to the aMED, HEI-2010, AHEI-2010, and DASH score in different quintile divisions (Q1-Q2-Q3-Q4-Q5) and the risk of osteoporotic fracture.	aMED: hip fracture: HR = 0.80 (95%CI 0.66–0.97) *** total fracture: HR = 1.01 (95%CI 0.95–1.07) *** (NS),
									HEI-2010: hip fracture: HR = 0.87 (95%CI 0.75–1.02) *** (NS) total fracture: HR = 0.98 (95%CI 0.93–1.02) *** (NS),
									AHEI-2010: hip fracture: HR = 0.94 (95%CI 0.80–1.09) *** (NS) total fracture: HR = 1.01 (95% CI 0.96–1.05) *** (NS),
									DASH: hip fracture: HR = 0.89 (95%CI 0.75–1.06) *** (NS) total fracture: HR = 0.98 (95%CI 0.94–1.03) *** (NS).
Fung et al., 2018 [39]	The United States	111,048, F (74,446) M (36,602)	(50–75)	Prospective cohort	No prescribed eating patterns	aMED score, AHEI-2010 score, DASH score	Hip fracture (self-reported)	Association between adherence to the aMED, AHEI-2010, and DASH score in different quintile divisions (Q1-Q2-Q3-Q4-Q5) and the risk of osteoporotic fracture in different genders.	aMED Women: hip fracture: HR = 0.96 (95%CI 0.81–1.12) **** (NS) Men: hip fracture: HR = 0.92 (95%CI 0.69–1.22) **** (NS)
									AHEI-2010 Women: hip fracture: HR = 0.87 (95%CI = 0.75–1.00) **** (NS) Men: hip fracture: HR = 0.88 (95%CI 0.67–1.17) **** (NS)
									DASH Women: hip fracture: HR = 0.95 (95%CI 0.815–1.11) **** (NS) Men: hip fracture: HR = 0.98 (95%CI 0.73–1.31) **** (NS).
Benetou et al., 2018 [41]	Europe and the USA	Total 131,241 F (110,459) M (20,782)	≥60	Prospective cohort	Mediterranean diet	mMED score	Hip fracture (diagnosed or reported at follow-up or recorded as cause of death)	Association between adherence to the mMED score in different tertile divisions (T1-T2-T3) and the risk of osteoporotic fracture in both genders.	Both genders: EPIC-Umea Sweden cohort: hip fracture: HR = 0.75 (95%CI 0.41–1.36) ****** (NS)
									NHS-USA cohort: hip fracture: HR = 1.02 (95%CI 0.91–1.15) ****** (NS)
									COSM-Sweden cohort: hip fracture: HR = 0.97 (95%CI 0.83–1.13) ****** (NS)
									SMC-Sweden cohort: hip fracture: HR = 0.91 (95% 0.82–1.03) ****** (NS).

* Adjusted for age, height, weight, disease or medication reducing BMD, smoking, intervention group, vitamin Ca and D supplementation, duration of hormone therapy and energy intake. ** Adjusted for age, age at menarche, age at menopause, BMI, physical activity, parity, duration of lactation, energy intake, sunlight exposure, smoking, supplement intake, and education. *** Adjusted for age, BMI, race/ethnicity, smoking status, physical activity, self-reported health, diabetes mellitus status, history of fracture at 55 years or older, physical function score, number of chronic medical conditions, number of psychoactive medications, duration of hormone therapy, bisphosphonates, calcitonin, and selective oestrogen receptor modulators. **** Adjusted for age, energy intake, BMI, height, smoking, leisure-time physical activity, postmenopausal hormone use (women), thiazides, Lasix, anti-inflammatory steroids, multivitamin supplements, calcium, retinol and vitamin D supplementation, intake of caffeine, sugar-sweetened beverages (Alternate Mediterranean Diet score only), and alcohol (Dietary Approaches to Stop Hypertension score only), and history of diabetes. ***** Adjusted for age, sex, BMI, height, diabetes prevalence, smoking status (and pack-years), physical exercise, educational level, living alone, total energy intake, energy-adjusted dietary intake of calcium, vitamin D and retinol, calcium or vitamin D supplementation, and Charlson weighted comorbidity index. ****** Adjusted for sex (in the EPIC-Elderly cohorts) and baseline characteristics of age, BMI, body height, physical activity at sports (not available in EPIC-Elderly Umea), smoking status, marital status, education level (not applicable in NHS-USA), history of comorbidity, and total energy intake.

### 3.3. Summary of ‘a Priori’ Dietary Scores

This review included six observational studies that examined a total of seven dietary scoring systems, including the BSD score, aMED score, HEI-2010 score, AHEI-2010 score, and two DASH and two mMED scores with slightly different scoring components and criteria. Specifically, different diet scores have different quantile divisions and corresponding score intervals, and the scores ranging from one to the highest reflect a gradual increase in adherence to dietary indexes. For scoring purposes, BSD scores are split into quartiles, aMED, HEI-2010, AHEI-2010 scores are divided into quintiles, mMED scores are separated into tertiles, and DASH scores are divided into tertiles and quintiles in the two studies, respectively. The scoring components and criteria are explained in Table 2.

From the dietary scoring systems mentioned above, two (DASH score and mMED score in Byberg et al.) have eight scoring components, three (BSD score, aMED score, mMED score in Benetou et al.) have nine scoring components, one (AHEI-2010 score) has eleven scoring components, and one (HEI-2010 score) has twelve scoring components. All diet scores capture the consumption of fruit, vegetables, grains, and meat products (red and processed meats). Except for the DASH and HEI-2010 scores, other dietary indexes include alcohol and fish consumption as part of the dietary assessment. However, these two dietary scores specifically assess sodium intake, while others do not limit its consumption. For dairy products, the BSD, DASH, and HEI-2010 score expressly specify low-fat milk/dairy as a scoring component, while mMED in Byberg et al. includes fermented dairy products, and mMED in Benetou et al. includes dairy products in the broad sense as a scoring component. Additionally, BSD, aMED, HEI-2010, and mMED in Benetou et al. scores assess the quality of dietary fat by calculating the fat ratio for dietary patterns, while AHEI-2010 split out fat types for scoring. Notably, compared to other dietary indexes, HEI-2010 focuses more on critical nutrients: protein, which includes the consumption of seafood, plant, and total protein foods.

### 3.4. The Effect of Diet Scores on BMD and Osteoporosis Diagnosis

Only two studies included in this scoping review evaluate the link between diet scores and BMD or osteoporosis (using dual X-ray absorptiometry (DXA)) [36,37]. The changes in BMD in older women corresponding to the different quantiles of the two diet scores are shown in Appendix B. Overall, there was no significant association between BMD (lumbar, femoral and total body) and BSD scores in the multivariable-adjusted analysis [36]. Similarly, the association between BMD (as a continuous variable) and DASH diet score tertile was not significant, but adherence to the highest tertile of DASH score was associated with a significantly lower risk of lumbar spine osteoporosis compared with the lowest tertile (HR = 0.28; 95%CI 0.09-0.88) [37]. Remarkably, comparing the difference in the effect of the two dietary patterns on BMD (based on diet scores) between recruits in the two study cohorts, older people following the DASH diet had lower mean BMD in the lumbar and femoral neck than those following the BSD.

### 3.5. The Influence of Diet Scores on Osteoporotic Fracture

In general, there were four studies that analysed the association between six diet scores and the risk of osteoporotic fracture (four studies investigated the association between diet scores and hip fracture risk, and one of these studies additionally investigated the relationship between diet score and total fracture risk). After data integration, 19 data sets (with region, gender, diet score and osteoporotic fracture site differences) are presented as a forest plot in Figure 2. According to the 95% confidence intervals listed in the forest plot below, a significant association between adherence to modified and alternative MD, respectively, and lower risk of osteoporotic fracture was shown in only two datasets: one from the Swedish cohort (including both genders), where better adherence to modified MD (assessed using mMED score) was associated with lower incidence of hip fracture (HR = 0.78 (95%CI 0.69–0.89)) [40], and one from the USA female cohort, where women with the highest adherence to the alternate MD (assessed using aMED score) had a lower incidence of hip fracture (HR = 0.80; 95%CI 0.66–0.97) [38]. In addition, the remaining data show that higher adherence to DASH, AHEI-2010, mMED2 and HEI-2010 scores do not correlate with hip fracture risk, and higher commitment to DASH, aMED, HEI-2010 and AHEI-2010 scores do not associate with total fracture risk. Notably, for the same dietary index (DASH, aMED, and AHEI-2010), higher adherence to each diet score may link to lower hip fracture risk in women than in men.

## 4. Discussion

This is the first scoping review focusing on the relationship of different alternative dietary patterns with osteoporosis and its associated fracture risk in older people. Six studies were included in the review. Specifically, higher adherence to the DASH diet was associated with a lower probability of developing osteoporosis in the lumbar spine in postmenopausal women in one small study, although adherence to DASH was not associated with risk of hip fracture in a larger study including men and women. Higher adherence to the AHEI-2010 and aMED scores may contribute to some reduction in hip fracture risk in older women. One study found that higher commitment to the mMED scores is beneficial in reducing hip fracture risk in older men and women, whereas another study found non-significant results.

These findings provide more possibilities that other healthy dietary patterns (other than the traditional Mediterranean diet) may also promote bone health and mitigate the risk of osteoporotic fractures and osteoporosis. In addition, as these four studies included individuals from non-Mediterranean countries, it can be inferred that closer adherence to the alternative or modified Mediterranean diet is also conducive to lowering the risk of hip fracture in non-Mediterranean populations. However, in the context of this review, a risk of bias assessment was not undertaken. As this is a scoping review, it does not aim to provide a definitive answer about the effectiveness of each diet but rather to investigate gaps in the research on alternative dietary patterns using an ‘a priori’ scoring system in relation to osteoporosis and fractures.

The comparison of results on bone health and osteoporosis-related outcomes among each diet score is split into three categories to analyse the effect of the dietary index on BMD, osteoporosis and osteoporotic fractures. Although there was no significant association between the BSD or DASH diet scores and BMD in older women, those who followed the BSD (in Finland) had a much higher mean BMD than those who followed the DASH diet (in Iran), which is inconsistent with previous findings in the literature [10,31,32]. Around half of the women who followed the BSD in Finland received additional calcium and vitamin D supplements. Additionally, the number of women recruited in Iran who followed the DASH diet was small (151), which might lead to bias (if the sample was not of sufficient size to detect a significant difference). In this study, authors found that a higher diet score had a strong link with reduced risk of osteoporosis in the lumbar spine but not in the femoral neck [37]. According to data from the Nutritional Health and Nutrition Examination Survey (NHANES), in people aged 50 or above, the prevalence of osteoporosis in the lumbar spine is 6%, while the prevalence of osteoporosis at the femoral neck is 5%; therefore, the overall prevalence in these two sites has little difference [43]. However, the prevalence per site can vary by age. The age range of the Iranian study population [37] was 50–85, and the mean age was 61 years; moreover, we know that this population of postmenopausal women have a higher risk of BMD loss in the lumbar spine (as opposed to older women being more prone to BMD loss in the hip) [44].

Regarding osteoporotic fracture analysis, two studies found that participants with higher adherence to aMED and mMED scores have a lower risk of hip fractures. However, results on aMED scores differed between two cohorts of older adults in the USA [38,39]. This discrepancy might be due to the large difference in sample size between the two studies, with the study by Fung recruiting ten times as many women as the one by Haring. Additionally, the Fung study specifically adjusted for the intake of multivitamin supplements. The lack of information on multivitamin supplementation in the other study might also have affected the results; in view of the difference in the variables the analysis was adjusted for between these studies, their results cannot be directly comparable. Regarding the results of mMED diet scores, in a Swedish cohort, a significant association was found between higher mMED scores and lower risk of hip fracture in both sexes [40]. In contrast, in another study that included four non-Mediterranean cohorts (one from the USA and three from Sweden), adherence to the MD based on the mMED score did not reduce the risk of hip fracture [41]. The difference between the two scoring systems is that mMED in Byberg et al. included the use of olive or rapeseed oil (%) as a component, while mMED in Benetou et al. assessed the fat ratio of monounsaturated and polyunsaturated fats to saturated fats. This modification was done to allow for the score to be applied in non-Mediterranean populations, in which consumption of olive oil is not traditional and thus, intake of monounsaturated lipids from olive oil is very low [41]. High consumption of olive oil, apart from its other recognised health benefits [45], has also been associated with a lower risk of osteoporotic fractures in the PREDIMED trial [46]. This study recruited 870 participants at high cardiovascular risk aged 50–85 years in Spain who were randomised to MD supplemented with extra-virgin olive oil, MD supplemented with nuts, or a low-fat diet. Participants in the highest tertile of extra-virgin olive oil consumption had a 51% lower risk of fractures compared to those in the lowest tertile after adjusting for potential confounders, although total and common olive oil consumption was not associated with fracture risk.

After analysing the results of the included studies about the association between each of the five dietary scores and the risk of hip and total osteoporotic fractures, we cannot make any firm conclusions regarding which diets are beneficial to help reduce fracture risk due to the heterogeneity of the included populations. It has been suggested that osteoporosis may be caused partly by consuming more acid precursors than base precursors [47]. The traditional MD comprises a higher intake of alkali-forming food categories (fruit and vegetables) that are beneficial to altering the acid-base balance of the diet and can prevent bone loss [47].

There are some limitations in this review that mainly have to do with the included studies. Based on the pre-defined age range of people aged 50 years and above (or an average age of 60 years), three articles were excluded. More specifically, the participants’ age range in these studies was between 47–77, 40–75, and 40–60, respectively [48,49,50]. The first study [48] mainly focused on postmenopausal women, while the remaining two [49,50] included middle-aged populations. As BMD in women generally declines significantly after menopause, some studies propose that identifying increased risk factors for low BMD in people under 50 could help provide earlier dietary intervention when the osteoporosis threshold is reached [51]. However, there is no specific bounded age range for postmenopausal women, and using menopause alone as an indicator might not be sufficient to estimate the impact of age on bone turnover. Similarly, the boundaries between middle-aged and older people are also not clearly delineated, which affects the summarization of the influence of various dietary patterns or diet scores on osteoporosis in the different age groups to a large extent.

In addition to age, in the six studies selected in this review, the study groups have ranged significantly from one hundred to slightly over one hundred thousand people, and the relatively small number of research groups may impact the outcomes of the analysis. It is well known that dietary assessments were carried out using self-reported data, which depended on participant accuracy of recall and thus may cause reporting errors. Hence, the insignificant results of BMD and osteoporosis in the two studies [36,37] might be attributable to the restricted analytical power caused by the small sample size.

By summarizing the findings and the shortcomings of the included papers, this review has the following suggestions for further research. Firstly, more observational studies with longer follow-up and various study groups from different races and areas are needed to validate the benefits and drawbacks of the specific dietary patterns to prove their applicability to the public. Secondly, future participant recruitment needs to meet certain criteria, for example, a sufficient sample size, appropriate age limits, definition of baseline health status and/or comorbidities that affect the incidence of osteoporotic fractures, as well as a prospective cohort design with sufficient follow-up to reduce bias. Thirdly, as the potential confounders are not always effectively controlled, and the multifactor adjustment has some differences between each article, it might be helpful to develop a consensus regarding confounders that need to be accounted for in future research on this topic. Fourthly, future research should not be restricted by the existing ‘a priori’ diet scores. More regional dietary indices can be created to help different populations mitigate the risk of osteoporosis more effectively based on the dietary characteristics of people in different regions. In addition, a future review can broaden the scope of comparison to compare the impact of diverse ‘a priori’ diet scoring systems and the ‘a posteriori’ diet scoring systems on osteoporosis or osteoporotic fractures in older people.

Although this is a scoping review (which means it does not include risk of bias assessment to determine the strength of the evidence), our findings have potential value in promoting the development of non-pharmaceutical approaches to prevent osteoporosis and fractures. After analysing each scoring system, it can be inferred that a higher consumption of fruit, vegetables, whole grains, legumes, fish and healthy oil (especially olive oil), as well as maintaining a healthy intake fat ratio (relatively high intake of MUFA and PUFA, and low intake of SFA) are beneficial for bone health. These useful components can be considered as practical dietary guidance in people aged above 50 to help improve or maintain bone health in the long term. These interventions should be first tested for effectiveness in large-scale studies (ideally in a randomised clinical trial design) before recommendations to the public could be made. However, diet alone is not enough to prevent osteoporosis and fractures, and there is already evidence about the effectiveness of physical activity, including resistance training for osteoporosis [52,53]. Exercise and diet have the potential to inhibit biomarkers, such as interleukins (IL-1, IL-6), C-reactive protein (CRP), and tumour necrosis factor–α (TNF–α), to reduce the risk of fractures and muscle deterioration in osteosarcopenia [54]. Therefore, the combination of a healthy diet and exercise is crucial and has great potential in improving osteoporosis outcomes.

## 5. Conclusions

According to the results of six observational studies included in this scoping review, for people aged above 50, there is some evidence that adherence to the DASH diet is associated with a lower risk of osteoporosis in the lumbar spine and that adherence to a modified Mediterranean diet, DASH and alternate healthy diet could reduce hip fractures. These findings could be promising in promoting alternative healthy dietary patterns to maintain bone health. There is a need to optimise existing dietary indices and improve their relevance for bone health.

## Figures and Tables

**Figure 1 nutrients-15-04255-f001:**
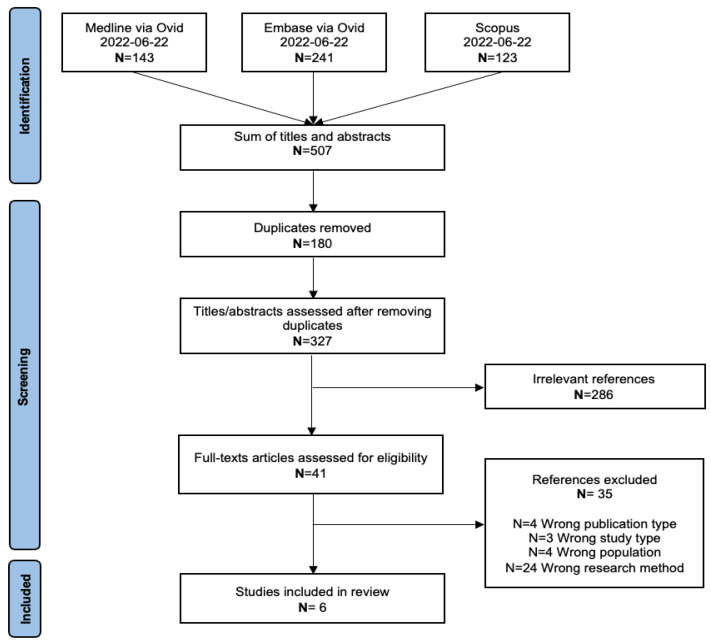
PRISMA Flowchart of study selection process.

**Figure 2 nutrients-15-04255-f002:**
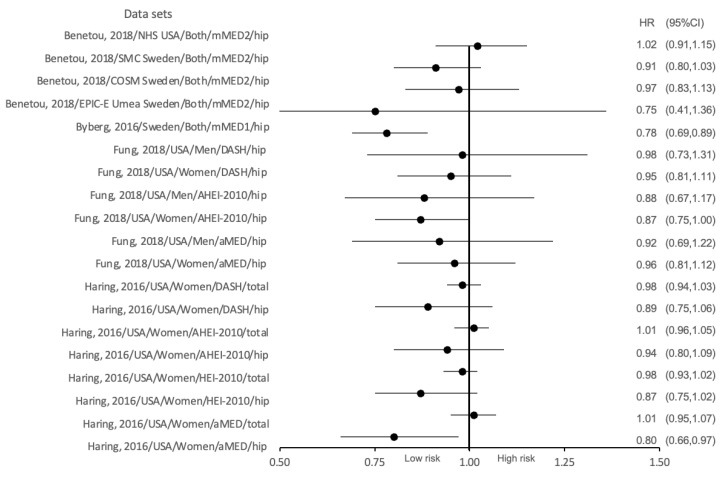
The association between adherence to different diet scores and hazard ratios (HRs) of osteoporotic fracture [38,39,40,41].

**Table 2 nutrients-15-04255-t002:** Summary of different scoring systems of 6 observational studies.

Scoring System		BSD	DASH		aMED	HEI-2010	AHEI-2010	mMED (Byberg et al. [40])	mMED (Benetou et al. [41])
Scoring component	1	Fruits and berries	Fruits		Fruits	Total vegetables	Whole fruit	Fruits and vegetables	Fruits
	2	Vegetables	Vegetables		Vegetables	Total fruit	Vegetables	Legumes and nuts	Vegetables
	3	Cereals	Whole grains		Whole grains	Whole fruit	Nuts and legumes	No refined or high fibre grains	Legumes
	4	Low-fat milk	Low-fat dairy		Fish	Seafood proteins	Whole grains	Fermented dairy products	Cereals
	5	Fish	Nuts and legumes		Nuts	Plant proteins	PUFAs	Fish	Fish
	6	Meat products	Red and processed meats		Legumes	Total protein foods	Long-chain ω−3 polyunsaturated fatty acids	Use of olive or rapeseed oil (%)	Meat
	7	Total fat	Sodium		Fat ratio	Whole grains	Red and processed meats	Red and processed meats	Dietary products
	8	Fat ratio	Sweetened beverages		Red and processed meats	Low-fat dairy	Sugar-sweetened beverages and fruit juice	Alcohol	Fat ratio
	9	Alcohol			Alcohol	Fatty acids ratio	Trans fat		Alcohol
	10					Refined grains	Sodium		
	11					Sodium	Alcohol		
	12					Empty calories			
	Total	9	8		9	12	11	8	9
Scoring criteria									
Quantile segmentation		Quartiles	Tertile ^a^	Quintile ^b^	Quintile	Quintile	Quintile	Tertile	Tertile
Detailed scores		Components (1–8) were scored according to sex-specific population consumption quartile points: the consumption of 1,2,3,4,5 and 8 was positively awarded for 0–3 points (0,1,2,3), while the scores of 6 and 7 were pointed vice versa. For 9, Men consume ≤ 20 g/d or women consume ≤ 10 g/d received 1 point; otherwise, received 0 points.	Components (1–8) were scored according to sex-specific population consumption quintile points: the consumption of 1–5 was awarded for 1–5 points (1,2,3,4,5), while the scores of 6–8 were pointed vice versa.		Components (1–8) were scored according to sex-specific population consumption median points: the consumption of 1–7 above the sex-specific median was awarded for 1 point, otherwise received 0, while the scores of 8 were pointed vice versa. For 9, Men consume 10–25 g/d or women consume 5–15 g/d received 1 point; otherwise, received 0 points.	Components (1–12) were scored according to sex-specific population consumption quintile points: the consumption of 1–6 was awarded for 0–5 points; the consumption of 7–11 was rewarded for 0–10 points; the consumption of 12 was rewarded for 0–20 points.	Components (1–11) were scored according to sex-specific population consumption quintile points: the component 1–11 was rewarded for 0–10 points.	Components (1–7) were scored according to sex-specific population consumption median points: the consumption of 1–6 above the sex-specific median was awarded for 1 point, while the scores of 7 were pointed vice versa. For 8, both genders consume 5–15 g/d 1 point; otherwise, received 0 points.	Components (1–7) were scored according to sex-specific population consumption median points: the consumption of 1,2,3,4,5 and 8 above the sex-specific median was awarded for 1 point, while the scores of 6–7 were pointed vice versa. For 9, Men consume 10–50 g/d or women consume 5–25 g/d received 1 point; otherwise, received 0 points.
Total scores		25	40	40	9	100	110	8	9
Score interval for different quantiles		Q1 ≤ 9	T1 10–22	Q1< 20	Q1 < 2	Q1 < 53	Q1 < 47	T1 0–2	T1 0–6
		Q2 10–13	T2 22–26	Q2 20–23	Q2 2–4	Q2 53–60	Q2 47–53	T2 3–5	T2 4–5
		Q3 14–15	T3 27–35	Q3 23–25	Q3 4–5	Q3 60–66	Q3 53–59	T3 6–8	T3 6–9
		Q4 ≥ 16		Q4 25–28	Q4 5–6	Q4 66–72	Q4 59–65		
				Q5 >28	Q5 > 6	Q5 > 72	Q5 > 65		
Notes		-8 = PUFA/(SFA + trans-fatty acids)			-7 = MUFA/SFA	-2 includes 100% fruit juice -3 includes all forms except juice -4 includes legumes (beans and peas) -9 = (PUFAs + MUFAs)/SFAs -13 includes energy from solid fats, added sugars, and any alcohol more than 13 g per 1000 kcal		-1 does not include fruit juice and potatoes	

For detailed scores and notes, numbers come from the corresponding scoring component’s serial number in each scoring system. -^a^ DASH score from Shahriarpour et al., 2020. -^b^ DASH score from Haring et al., 2016 and Fung et al., 2018. –PUFA = polyunsaturated fatty acids, SFA = saturated fatty acids, MUFA = monounsaturated fatty acids.

## Data Availability

No new data were created or analyzed in this study. Data sharing is not applicable to this article.

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
