# Peer review of "Association of Alternative Dietary Patterns with Osteoporosis and Fracture Risk in Older People: A Scoping Review"

_nutrients, 2023, doi:10.3390/nu15194255_

Round 1

Reviewer 1 Report

Many thanks to the authors for having presented a so interesting scoping review about “Association of Alternative Dietary Patterns with Osteoporosis 2 and Fracture Risk in Older People: a Scoping Review”.

Before resubmitting the revision version of the article, please read the editorial rules carefully, and check other editorial aspects, such as: text alignment (lacking), text justification at the head (lacking), etc. The language is no good, hence the manuscript needs to be corrected by a person of English mother tongue.

Plagiarism

Detected plagiarism:  12% (they should be < 15%). Fine!

Title and Abstract

The abstract is not well structured, although it contains the main results of the study, because the manuscript does not reflect the Strobe Statement-Checklist for cohort studies. Please read these guidelines for articles (also in Italian) before resubmitting the revision version. Hence, make sure that the different paragraphs are divided properly.

In the conclusion of the Abstract the correlation of the risk of hip fracture is described exclusively with mMD and aMD, but not with DASH. Furthermore, the correlation of the first two diets with the improvement of BMD is not mentioned. Please be more complete in this section.

Key words

Please provide them in alphabetic order.

Background

Although the introduction is quite well structured it is too short and focused on general populations rather than on elderly patients. Considering that menopause contributes to the increase in bone mass loss (therefore to the reduction of BMD) and to the risk of osteoporosis, I recommend also taking this variability into consideration and evaluating the importance of these alternative dietary regimes in the peri- and post-menopausal period for give greater accuracy to the study. Please add some references about this item including: ·      “ The Mediterranean Diet in Osteoporosis Prevention: An Insight in a Peri- and Post- Menopausal Population. Sara Quattrini, Barbara Pampaloni, Giorgio Gronchi, Francesca Giusti, Maria Luisa Brandi. Nutrients. 2021 Feb 6;13(2):531. doi: 10.3390/nu13020531.” Please, on line 95 explain the importance of the DII, explaining its health linkages. Finally, line 50: there is a gap about the difficult management of these fractures during the hospitalization. Between epidemiology and possible prevention describe in the introduction section, add these aspects.

Methods

This section contains enough information to understand and possibly repeat the study. However, it is not well structured, and it does not reflect the Strobe Statement-Checklist for cohort studies. As it contains a lot of results, they have to move into Results section, i.e. clinical notes of the hospital.

Statistical analysis

Please provide who performed the analysis and describe the programs used for multivariable ones.

Results

The results presented are quite complete, reflecting the MM section. 

The results are reproducible and reflective of clinical expectations.

Discussion

The length and content of the discussion communicates the main information of the paper. However, it recognizes the limitations of the manuscript, due to lack of multivitamin supplementation control, which affected the results and led to heterogeneity in participants’ nutritional levels, as described on lines 331 and 332.

Conclusions

The conclusions reflect and refer to the results of the study. Please delete… “Additionally, more studies need to be carried out in 423 different regions to validate the association between various regional dietary patterns and 424 the risk of osteoporosis and fractures in older adults.”, as it is not objective of our study.

References

The references are up to date but delate those before 2010 if not essential and replace them with newer ones and those suggested previously. 

Competing interest

The authors' competing interests do not appear created a bias in the reporting of the results and conclusions.

Tables and Figures

The quality of tables and figures are appropriate to transmit the main information of the paper. However, provide a proper Tables and Figure legends section.

The absence of tables that correlate the effect of MD and alternative diets with pro-inflammatory values ​​makes the objective of the study less effective. I recommend learning more about this section.

Recommendations to Editors

Paper requires some major and multiple minor revisions.

Plagiarism

Detected plagiarism:  12% (they should be < 15%). 

Moderate editing of English language required.

Reviewer 2 Report

I consider this article very relevant on the topic: Association of Alternative Dietary Patterns with Osteoporosis and Fracture Risk in Older People: a Scoping Review, given that scientific evidence on this topic is scarce. Aging is related to physiological changes that affect the bioavailability of nutrients that can favor the development of osteoporosis. This osteometabolic disorder is characterized by a reduction in bone mass and deterioration of the microarchitecture of bone tissue, promoting bone fragility and, consequently, increasing the predisposition to fractures, especially in women after menopause. As the etiology of bone mass loss is complex and multifactorial, nutrition plays a crucial role in the prevention and treatment of osteoporosis. We know that adequate consumption of nutrients involved in bone metabolism, at all periods of the life cycle, can prevent or reduce the incidence of osteoporosis. Therefore, increasing of other dietary patterns other than the traditional Mediterranean pattern will certainly be an added value.

 I therefore make some suggestions that I believe could be a contribution to clarifying and enriching the research:I would like to have seen all the alternative dietary patterns studied addressed and characterized in the methodology: Baltic Sea Diet (BSD), Modified/alternative Mediterranean Diet, Dietary Approaches to Stop Hypertension (DASH)).

I consider that tables 1 and 2 are not clear, they are confusing, I suggest another presentation in which there is a clear understanding of the data presented.
